# Postoperative Radiotherapy of Prostate Cancer: Adjuvant versus Early Salvage

**DOI:** 10.3390/biomedicines10092256

**Published:** 2022-09-12

**Authors:** Daniel Wegener, Daniel M. Aebersold, Marc-Oliver Grimm, Peter Hammerer, Michael Froehner, Markus Graefen, Dirk Boehmer, Daniel Zips, Thomas Wiegel

**Affiliations:** 1Department of Radiation Oncology, University Hospital Tuebingen, 72076 Tuebingen, Germany; 2Department of Radiation Oncology, Inselspital Bern University Hospital, University of Bern, 3012 Bern, Switzerland; 3Department of Urology, Jena University Hospital, 07743 Jena, Germany; 4Department of Urology, University Hospital Braunschweig, 38106 Braunschweig, Germany; 5Department of Urology, Zeisigwaldkliniken Bethanien Chemnitz, 09130 Chemnitz, Germany; 6Martini Clinic, University Medical Center Hamburg-Eppendorf, 20251 Hamburg, Germany; 7Department of Radiation Oncology, Charité University Medicine Berlin, 10117 Berlin, Germany; 8German Cancer Consortium (DKTK), Partner Site Tuebingen, German Cancer Research Center (DKFZ), 69120 Heidelberg, Germany; 9Department of Radiation Oncology, University Hospital Ulm, 89081 Ulm, Germany

**Keywords:** prostate cancer, radiotherapy, postoperative radiotherapy, adjuvant radiotherapy, salvage radiotherapy

## Abstract

Results of three randomized clinical trials (RCTs) comparing adjuvant radiotherapy (ART) and early salvage radiotherapy (eSRT) of prostate carcinoma and a subsequent meta-analysis of the individual patient data from these RCTs were recently published. The results suggest that early eSRT is as effective and potentially less toxic than ART. Therefore, eSRT should be considered the standard of care. However, due to limitations in the RCTs, ART remains a valid treatment option in patients with the combination of high-risk features such as Gleason Score (GS) 8–10, positive surgical margins (R1) and pathological T-stage 3 or 4 (pT3/4). This article provides a critical appraisal of the RCTs and the rationale for recommendations adopted in the current national guidelines regarding patients with high-risk features after radical prostatectomy (RP): ART should be offered in case of pT3/pT4 and R1 and Gleason Score 8–10; ART can be offered in case of pT3/pT4 and R0 and Gleason Score 8–10 as well as in case of multifocal R1 (including pT2) and Gleason Score 8–10. In any case, the alternative treatment option of eSRT in case of rising PSA should be discussed with the patient.

## 1. Introduction

Current international and national guidelines for post-prostatectomy patients with adequate PSA response (<0.1 ng/mL) and with high risk of recurrence recommend either adjuvant radiotherapy (ART) or early salvage radiotherapy (eSRT) [1]. This recommendation is based on previous randomized clinical trials (RCTs) [2,3,4] which demonstrated a reduced risk of biochemical recurrence of approximately 20% in patients with high risk of recurrence after radical prostatectomy (RP, i.e., positive surgical margins (R+), pT3/4 disease, GS 8–10). Follow-up data for survival endpoints and toxicity reached about 10 years in these studies. However, these RCTs compared ART vs. surveillance (based on current data) and late use of SRT related to imprecise PSA monitoring. In clinical practice, ART was not always routinely administered in high-risk patients. Presumably, this had multiple reasons such as decision against ART by patient or treating physician related to the risk of overtreatment or concerns about toxicity, especially in case of persistent urological symptoms after RP. In addition, clinical experience (but not prospective trials) with eSRT, revealed high rates of biochemical control and no obvious excess toxicity which led to the introduction of eSRT in clinical practice despite the lack of RCTs comparing ART and eSRT. In 2020, three RCTs and one preplanned meta-analysis were published [5,6,7,8] (Table 1). 

## 2. Methods

A critical analysis of the recent RCTs comparing ART with eSRT and their meta-analysis was performed. The authors evaluated whether the recommendations following these studies could be generalized and should be adopted for patients featuring high-risk features (R+, pT3/4, GS 8–10). Results of this assessment were incorporated into the German national S3 guideline. The workflow is demonstrated in Figure 1.

## 3. Results

The “Raves” study by Kneebone et al. [5] was conducted between 2009 and 2015 in New Zealand and Australia and aimed to randomize 470 patients to either receive ART 4–6 months after RP or salvage RT of 64 Gy in 32 fractions, triggered if post-RP PSA breached 0.2 ng/mL. Included were patients with R+, notably possibly including pT2 patients, extraprostatic extension and/or seminal vesicle invasion. Postoperative PSA needed to be below 0.1 ng/mL. This study’s power calculations and intended sample size were based on an assumed 74% freedom from biochemical failure rate 5 years after treatment (FFbF, defined as PSA > 0.4 ng/mL) in the adjuvant group and would consider salvage RT non-inferior if FFbF was at most 10% lower than that. However, in 2015 the trial closed for recruitment after 333 included patients due to “too little events” occurring and reaching adequate power for the primary endpoint was deemed infeasible. 

Results at the cutoff date for analysis in 2018 showed after a median follow-up (FU) of 6.1 years that the aforementioned FFbF did not differ between the two groups (86% vs. 87%). A total of 84/167 (50.3%) patients of the eSRT arm had a biochemical relapse and 80/167 (47.9%) had undergone eSRT at that point. In the intend-to-treat analysis, grade 2 or worse genitourinary (GU) toxicity, was lower in the eSRT group compared to the ART group (54% (90/167) compared to 70% (116/166)). Gastrointestinal (GI) grade 2+ toxicity was similar: eSRT at 10% (*n* = 16) and ART at 14% (*n* = 24).

The authors conclude that eSRT is to be considered a new standard of care.

The French “GETUG-AFU 17” trial by Sargos et al. [6] included 424 patients between 2008 and 2016 with pT3/4, pNx or pN0 carcinomas and R+. RT dose was 66 Gy in 33 fractions in both groups and RT of the pelvic lymph nodes was optional (46 Gy in 23 fractions). All patients received androgen deprivation therapy (ADT) consisting of 6 months of triptorelin. The primary outcome was event-free survival (EFS) and this study was designed to show a 10% increase in the 5-year EFS rate from 60% with eSRT to 70% with ART. To reach this endpoint, 718 patients were required yet this study was also closed prematurely in 2016 due to unexpectedly low event rates (only *n* = 12 (5%) of the expected 242 events had been reported thus far) and slow accrual. At that point, 54% (*n* = 115 patients) of the eSRT group had received RT (*n* = 205 in ART group). Median PSA level at the time of eSRT initialization was 0.24 ng/mL.

Results at the cutoff date and with a median FU of 75 months showed no significant difference in EFS between the two groups (92% vs. 90%). In the intend-to-treat analysis, there was however a significant difference in acute GI grade 2+ and GU grade 2+ toxicity in favor of eSRT. Additionally, there was a difference in GU toxicity grade 2+ (ART 27% (58/212) vs. eSRT 7% (14/212)) and in erectile dysfunction (ART 28% (60/212) vs. eSRT 8% (17/212)).

The authors discuss that eSRT could spare patients from toxicity and overtreatment.

The “RADICALSRT” trial from Parker et al. [7] was conducted in Canada, Denmark, Ireland and the United Kingdom. This study randomized 1.396 patients between 2007 and 2016 to receive either ART or eSRT. Inclusion criteria were either pT3/4, Gleason Score 7–10, R+ (pT2 possible) or initial PSA > 10 ng/mL. Originally, about 2600 patients were planned to be included to have 80% power to detect an improvement in disease-specific survival (DSS) from 70% to 75% or 90% power to detect an improvement from 80% to 85% at a two-sided 5% significance level. The primary endpoint was changed to freedom from distant metastases (FFdM) in 2011 because this study also recorded fewer events as initially planned. This study chose post-RP PSA of 0.2 ng/mL or less as a cutoff. RT was 66 Gy in 33 fractions or 52.5 Gy in 20 fractions in both groups and RT of pelvic lymph nodes was optional. Median PSA level at the time of eSRT initialization was 0.2 ng/mL. In the ART group, 24% of patients (154/649) and 27% in the eSRT group (61/228 patients) were additionally randomized to also receive ADT either for 6 or 24 months (either LHRH analogue or Bicalutamide 150 mg) or additional ADT was administered according to clinical judgement. 

Results at the cutoff date in 2019 after a median FU of 4.9 years showed that the primary endpoint FFdM could not be adequately evaluated due to low event rates. Rates for 5-year FFbF were 85% for ART vs. 88% for eSRT (*p* = 0.56). 647 (93%) of the ART group and 223 (32%) of the eSRT group had received RT at that point. In the intention-to-treat analysis comparing 696 patients in both groups, significantly higher rates of RTOG GU and GI toxicity items were reported for ART as well as higher rates of self-reported urinary- and feacal incontinence at one year (but not five years) for ART.

The authors conclude that ART posed a higher risk, especially for GU toxicity while no oncological benefit was seen. Therefore, eSRT should be regarded as the standard of care.

Vale et al. [8] published a meta-analysis of the aforementioned RTCs. This preplanned analysis was organized with the workgroups of the three RCTs in 2014 (ARTISTIC collaboration) before any study results were known and aimed to further elaborate on the comparison of ART and eSRT, especially since the event rates in all three mentioned RCTs were lower than initially expected and accrual rates for each primary endpoint could not be met. 2153 (ART *n* = 1075, eSRT *n* = 1078) patients were included. Out of the 1078 eSRT patients, 421 (39.1%) had received eSRT. Median follow-up was 60–78 months. Based on 270 events, no advantage of ART over eSRT was found (EFS 89% vs. 88%). No preplanned subgroup could be identified, for which ART was significantly beneficial.

The authors conclude that eSRT should be regarded as the standard of care.

## 4. Discussion

These four publications represent highly relevant evidence for the timing of post-RP radiotherapy. However, a number of limitations need critical discussion and careful consideration regarding the optimal choice of postoperative radiotherapy. 

### 4.1. Follow-Up Time Not Sufficient to Conclude on Oncological Outcome

For each of the studies FU (mean 60–78 months) is rather short for a comparison of oncological outcomes of RT after RP. Several previous RCTs (comparing ART to surveillance) could demonstrate that a significant (further) increase in the difference of oncological outcome parameters occurs over time. Hackman et al. found that the difference in biochemical recurrence free survival between post-RP ART or surveillance further increased after 5 years [9]. Similarly, the difference in OS further increased over time in the RCTs of Thompson et al. and so did FFbF in the RCT of Wiegel et al. [2,4].

The low rate of events and slow accrual led to the premature termination of all three RTCs. In the ARTISTIC meta-analysis based on 270 events (136 in eSRT group), an adapted 88% baseline survival at 5 years was assumed and 120 events in the eSRT arm were predicted (endpoint EFS, Table 1). A power of 90% to detect a 5% difference in EFS was successfully reached. However, since only 39.1% of patients in the eSRT arm (so far) received RT, it is possible that further events influence oncological outcome (as well as toxicity). 

### 4.2. Immortal Time Bias

The apparent oncological equivalence of eSRT and ART could be confounded by an immortal time bias. The aforementioned studies allowed for eSRT to start within four months after exceeding the PSA threshold (0.1 ng/mL in RADICALS-RT or 0.2 ng/mL). Then followed eSRT of eight weeks and up to three more months might have passed until the first post-RT PSA value was obtained. Therefore, for an extended period of time, patients receiving eSRT were systematically unavailable for triggering the PSA-cutoff (0.4 ng/mL) defining an event in the mentioned RTCs. 

This “immortal time bias” was first discussed by Tilki and D’Amico [10,11] and seems most relevant for patients with a high risk for a rapidly rising PSA, which includes GS 8–10, pT3/4 and especially a combination of those factors. Therefore, more mature data including more events are required to finally answer this relevant question.

### 4.3. High-Risk Subgroup Underpowered

A large proportion of patients in the meta-analysis do not feature risk factors that would currently warrant ART (GS 8–10 and pT3/4 characteristics, especially in combination). Especially in the RADICALS-RT trial, which contributed roughly two thirds of all patients for the meta-analysis, inclusion criteria allowed patients with little risk for recurrence to be randomized. In this study, 56% of patients had GS 7a or lower and only 17% had GS 8–10. 339 (24%) of patients had a pT2 carcinoma. Only 37 patients (2.7%) of both arms featured GS 8+ and pT3b/4 and R+. A combination of two risk factors (GS 8+ and R+, pT3b/4 and R+ and GS 8+ and pT3b/4) was given in 96 (6.9%), 122 (8.7%) and 23 patients (1.6%) (see Supplementary Figures S2 and S3 of Parker et al. [7]). This consequently means for the meta-analysis that >20% of patients had a pT2 carcinoma and only 20% of patients had pT3b. Only 16% of patients had GS 8–10. Additionally, no reference pathology was performed in the mentioned RCTs, which was shown to have a relevant impact especially on the Gleason Score [12]. In today’s practice, a large proportion of patients in the analysis would not be considered for ART (as today only the high-risk subgroup). It should be expected that even with longer follow-up, the three trials and the meta-analysis might be underpowered to sufficiently defend the statement that this high-risk subgroup does not benefit from ART.

These first three points of criticism were recently addressed in a retrospective analysis performed by Tilki et al., who evaluated the impact of ART vs. eSRT on overall mortality in over 26,000 patients with a median FU of 8.16 years. Especially in the high-risk group of GS 8–10 and pT3/4 (or GS 8–10 and pN1), a significantly reduced risk for all-cause mortality was found when ART was performed compared to eSRT [11]. Therefore, despite the limitations inherent in the retrospective design, this study appears to further support ART in patients with multiple high-risk features.

### 4.4. Impact of ADT on Trial Results

ADT for 6 months was administered to all patients in both arms in the GETUG-AFU 17 trial and several patients in the RADICALS-RT trial (ADT of six months in 90 patients (13.9%) and of 24 months in 45 patients (6.9%) of the ART arm and of six months in 33 patients (14.4%) and of 24 months in 13 patients (5.7%) who underwent RT in the eSRT arm, respectively). In the RADICALS-RT trial, ADT was also administered to some patients according to clinical judgement (*n* = 19 in ART am, *n* = 15 in eSRT arm), i.e., most likely to high-risk patients. ADT can delay time to progression and possibly also influence curation rates of ART (in the GETUG study only) or eSRT (in GETUG-AFU 17–and in RADICALS-RT). Possibly, the addition of ADT has led to fewer events especially in the eSRT groups of those studies (as in the mean, eSRT must have been administered several months later than ART and ADT might still be in effect for the remainder of the FU), creating a bias. This could further influence BFS rates and raises the question of the reliability of EFS as an endpoint.

### 4.5. Toxicity Possibly Just Delayed

Another point of criticism, also voiced by Tilki and D’Amico [10], is the “immortal time bias” on (RT related) toxicity scoring: Since toxicity in the eSRT arm could only increase after eSRT and since eSRT started months or years later than ART (data from GETUG-AFU 17: mean difference ca. 22 months), for a large period of the FU period, many patients in the eSRT group were locked out from an RT related toxicity increase. In other words, many patients received ART in the RCTs, who according to current standards would not be selected for ART. The toxicity reported by all these patients was compared to the toxicity of the subgroup of patients who had received eSRT (39.1%) in relation to all patients of the eSRT study arms and then revealed a higher toxicity for ART in the “intend to treat” analysis. When directly comparing the toxicity of those patients of both arms who actually received RT (“as treated”), ART was less toxic. Of course, this “intend-to treat” analysis is nevertheless correct and still adds valuable data as toxicity in the “interval” between RP and a possible eSRT is also highly relevant, but since the authors concluded that ART is to be regarded as more toxic than eSRT, this bias is pointed out. Therefore, it remains questionable whether the toxicity of ART is indeed increased over eSRT or simply “postponed” (which would still be advantageous if oncological outcomes are equal). 

### 4.6. RT Technique (IMRT vs. 3DRT) Might Have Influenced Outcomes

As previously mentioned, eSRT in the mean performed months to years after ART. Within the trials period (2007 to 2016), technological advanced such as IMRT became more widespread and replaced 3DRT in postoperative prostate cancer treatment. Evidence is found in GETUG-AFU 17 where 30% of ART was performed as IMRT compared to 47% of eSRT. Presumably, future patients in the eSRT arm will also mostly receive IMRT, further increasing this potential bias [13]. 

### 4.7. Heterogeneity and Inconsistencies in Radiation Dose and Target Volume Definition among the Three RCTs in the Meta-Analysis

The radiotherapy doses vary between the three RCTs (Raves trial: 64 Gy in both groups, RADICALS-RT and GETUG-AFU 17: 66 Gy in both groups, the latter two also partially included pelvic lymph nodes in the RT (RADICALS-RT: 7% in ART arm, 3% in eSRT arm; GETUG-AFU 17: 18% in ART arm, 24% in eSRT arm, Table 1). This difference in RT treatment dose and target volume poses a further risk of bias both for outcome and toxicity.

### 4.8. Radiation Dose and Fractionation Not Considered Evidence-Based Standard of Care

Adjuvant radiation doses in the studies were 64 and 66 Gy, i.e., higher than the currently accepted standard dose of 60 Gy, potentially leading to more toxicity. Hypofractionated 52.5 Gy in 20 fractions, as received by 29% (*n* = 258) of patients in the RADICALS-RT trial, appear not to be an established evidence-based standard alternative to conventional fractionation and with a 2 Gy equivalent dose of 60.7 Gy (a/b = 2 Gy) introduce another bias and source of heterogeneity complicating the interpretation and use in clinical practice.

### 4.9. ART Was Performed within Less Than 6 Months after Surgery

International guidelines (ASTRO, NCCN) would consider ART up to one year after surgery. This might negatively influence the toxicity risk in the ART arm.

### 4.10. Potential Inclusion of Patients with Persisting PSA after RP

In the RADICALS-RT trial, which contributed 64% of patients for the meta-analysis, cutoff for inclusion was a post-RP PSA of 0.2 ng/mL or less. Current guidelines [1] would consider postoperative PSA levels above 0.1 ng/mL as a PSA persistence with a potentially worse prognosis and would advise eSRT. Data on how many patients of RADICALS-RT fell into this category are not presented, therefore this potential bias is pointed out.

### 4.11. Risk of Delayed Referral to SRT

The perception among radiation oncologists and urologists that SRT is equivalent to ART might lead to a delay of referral to SRT with higher PSA and subsequently worse prognosis [14]. It is important to stress that the presumed equivalence is for early SRT (hazard ratio for biochemical failure of 1.41 for PSA before SRT of ≤0.2 ng/mL vs. >0.2–0.49 ng/mL in a recent study from Campbell et al.) [15]. This caveat holds especially for high-risk features such as GS 8–10 [16]. In addition, the use of PSMA-PET in the management of postoperative PSA failure was not part of the studies. Thus, there is no evidence from the RCTs to delay curative SRT until PSMA turns positive. Notably, this last argument is of general nature and not in the context of the recent RCTs.

## 5. Conclusions

No doubt, the recent trials tremendously contribute to high level evidence for the optimal care of patients and impact decision making for the majority of patients towards eSRT. These new data greatly improve patient selection towards eSRT in patients without multiple high-risk features and led to a reduction in “unnecessary” adjuvant treatments. However, given the above-mentioned limitations, several of which having also been voiced by the authors of the RCTs and the meta-analysis, the interdisciplinary working group within the German S3 guideline consortium has agreed upon recommendations for patient counselling in specific clinical scenarios where ART still appears to be a valid treatment option (Figure 2). These subgroups (no PSA persistence, nodal negative) include:

ART should be offered in case of pT3/pT4 and R1 and Gleason Score 8–10, ART can be offered in case of pT3/pT4 and R0 and Gleason Score 8–10, ART can be offered in case of multifocal R1 and Gleason Score 8–10 and pT2. In any case, the alternative of eSRT in case of rising PSA should be discussed with the patient. 

Further clinical data on the optimal management of patients after RP appear to be necessary. For example, on the optimal use of ADT (e.g., RADICALS-HD), on novel radiotherapy technologies, on role of hypofractionation, on the use of PET-CT and on the use of biomarkers for better risk stratification.

## Figures and Tables

**Figure 1 biomedicines-10-02256-f001:**
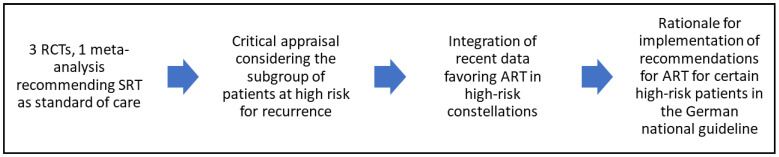
Flowchart of the procedure in this manuscript. RCT = randomized controlled trial; SRT = early salvage radiotherapy; ART = adjuvant radiotherapy.

**Figure 2 biomedicines-10-02256-f002:**
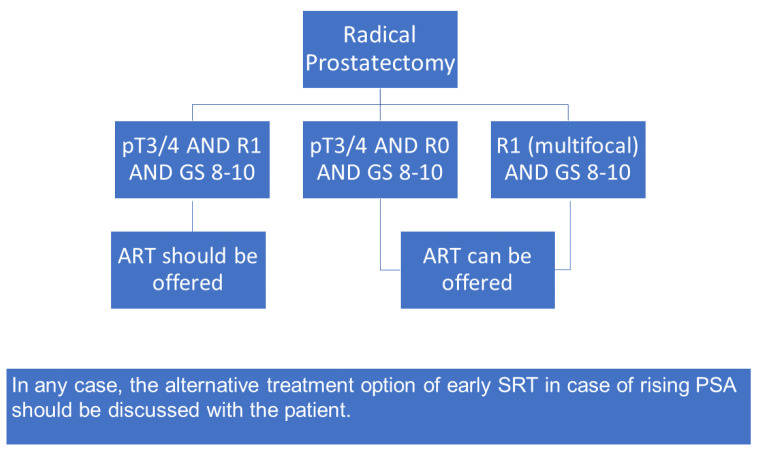
Organigram to aid decision making for patients with high-risk features. GS = Gleason Score. ART = adjuvant radiotherapy; SRT = early salvage radiotherapy; PSA = prostate specific antigen. Noteworthy, this organigram depicts the workflow in case of pN0 and a post-operative PSA-value of <0.1 ng/mL.

**Table 1 biomedicines-10-02256-t001:** Comparison of selected study characteristics and oncological and toxicity results of the RCTs RADICALS-RT, GETUG-AFU 17 and RAVES and the ARTISTIC meta-analysis.

	RADICALS-RT	GETUG-AFU 17	RAVES	ARTISTIC Meta-Analysis
Trial design	Superiority (two-sided, 80% power to detect increase from 90% to 95% 10-year FFDM, α = 5%)	Superiority of ART (one-sided, 80% power to detect increase of 10% 5-year EFS for ART over SRT, α = 5%)	Non-inferiority of SRT (one-sided, 80% power to detect 5-year FFbP difference of <10% for SRT, α = 5%)	Superiority (two-sided, 90% power to detect a 5% difference in 5-year EFS, 99% power to detect 10% difference)
Primary Outcome	FFDM	EFS	FFbP	EFS
Accrual period	2007–2016	2008–2016	2009–2015	
Planned accrual (*n*)	2600	718	470	(>120 events per group)
Actual inclusion (*n*, %)	1396 (53.7%)	424 (59.1%)	333 (70.9%)	2153 (events: 138 SRT, 126 ART)
Main inclusion criteria	One or more of: pT3/pT4, R+, Gleason Score 7–10, PSA > 10 ng/mL (including pT2)	pT3/pT4a and R+	pT3/4 or R+ (including pT2)	
RT dose for ART and eSRT	66 Gy in 33 Fx both groups (61%) or 52.5 Gy in 20 Fx (29%)	66 Gy in 33 Fx both groups (lymph nodes median 46 Gy)	64 Gy in 32 Fx both groups	
RT field	Prostate bed, additionally pelvic lymph nodes in 7% of ART arm and 3 % of SRT arm	Prostate bed, additionally pelvic lymph nodes in 18% of ART arm and 24 % of SRT arm	Prostate bed	
ADT	Randomized to 6 or 24 months of LHRH- analogue or Bicalutamid (RADICALS-HT double randomization) or as clinically indicated for 24% of ART arm and 27% of SRT arm.	All patients: 6 months of triptoreline	none	
ART timing	≤6 months after RP	≤6 months after RP	≤6 months after RP	
Trigger for eSRT	PSA > 0.1 ng/mL and rising or 3 × consecutive rising below 0.1 ng/mL	PSA > 0.2 ng/mL	PSA > 0.2 ng/mL	
eSRT timing	≤2 months of trigger PSA	ASAP after trigger PSA, before PSA of 1 ng/mL	≤4 months of trigger PSA	
Number of patients who received SRT	228/699 (32.6%)	115/212 (54.2%)	84/167 (50.3%)	421/1078 (39.1%) *
Median follow-up (months)	60	75	78	60–78 months
Main result (ART vs. eSRT)	5-year FFbP: 85% vs. 88%	5-year EFS: 92% vs. 90%	5-year FFbP: 86% vs. 87%.	5-year EFS: 89% vs. 88%
GU toxicity (ART vs. eSRT)	G3+: 9% (54/599) vs. 5.1% (32/621) #	G2+: 59% (125/212) vs. 22% (46/212)	G2+: 70% (116/166) vs. 54% (90/167)	
GI toxicity (ART vs. eSRT)	G3+: 2% (12/599) vs. 0.5% (3/621) ##	G2+: 8% (17/212) vs. 5% (11/212)	G2+: 14% (24/116) vs. 10% (16/167)	
% of IMRT (ART vs. eSRT)		30% vs. 47%		

Sources: table inspired by Vale et al., Tables 1 and 3. RCT = randomized clinical trial, FFDM = freedom from distant metastases, EFS = event free survival, FFbP = freedom from biochemical progression, ART = adjuvant radiotherapy, eSRT = early salvage radiotherapy, ECE = extracapsular extension, RP = radical prostatectomy, Fx = fractions, ADT = androgen deprivation therapy, LHRH = luteinizing hormone releasing hormone, GU = genitourinary, GI = gastrointestinal, IMRT = intensity modulated radiotherapy. * Numbers taken from each publication. Difference in sum of all trials compared to meta-analysis most likely due to patients who triggered PSA threshold for SRT but did not yet receive SRT and entered statistical calculations inconsistently. # Items cystitis G3+, hematuria G3+ and urethral stricture G3+ summed up. ## Items diarrhea G3+ and proctitis G3+ summed up.

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
