# Peer review of "Postoperative Radiotherapy of Prostate Cancer: Adjuvant versus Early Salvage"

_biomedicines, 2022, doi:10.3390/biomedicines10092256_

Round 1

Reviewer 1 Report

In this material the authors discuss one of the most relevant/important uro-oncology issue, the application of adjuvant/early salvage RT in the postoperative setting of prostate cancer. The bases of this manuscript are 3 recently published RCTs and a related meta-analysis, which works suggest the absolute omission of (early) adjuvant RT. The authors debate this general ascertainment and provide a kind of a „critical appraisal” of this statement. This approach, the criticism of the findings above, and the particular analysis of the biological/medical/statistical factors in the background is really important to understand/optimize the best care of our patients. To summarize the reviewer’s opinion, the material is worthy to consider for publication. However the reviewer has some notifications/concerns/recommendations.

1, It would be better to use consequently „early salvage RT” (eSRT) alongside the material to distinguish it from the other forms of salvage RT.

2, Consider the use of „postoperative” (e.g. in the title) since sometimes (e)SRT is carried out 2-3 years after the surgery intervention.

3, The ARTISTIC meta-analysis specified some weaknesses of the work as well, e.g. the short follow-up periods, the low number of events, the incalculable effect of endocrine treatment, the overrated value of PSA follow up etc. It would be worthy to mention it.

4, The ARTISTIC meta-analysis excluded the obligate effect of different prognostic factors, like basic PSA level, Gleason score, R state, vesicle seminalis infiltration, CAPRA score etc. It would be interesting to interpret it.

5, It is recommended to mention the significance of CAPRA score and Decipher molecular assay in the Discussion/Conclusion part.

6, The 3 RCT differed from each other considering inclusion criteria as well, e.g. in the obligatory number of risk factors (e.g. GETUG-AFU 17 required 3 risk factors, and interestingly in this study there was some trend in favor of ART, see ARTISTIC manuscript). It would be interesting to interpret it.

7, Abstract, lines 27-28. It is recommended to shorten/close up the „can be offered” categories.  

8, Line 41: The PSA monitoring was really „inadequate” in these studies?

9, Comments: „Immortal time bias„ could modify the results only some patients. Some delay of referral to SRT (line 261) likely has only minimal general effect as well.

10, It is interesting to discuss the role of PSMA-PET in this setting.   

Author Response

Reviewer #1:

In this material the authors discuss one of the most relevant/important uro-oncology issue, the application of adjuvant/early salvage RT in the postoperative setting of prostate cancer. The bases of this manuscript are 3 recently published RCTs and a related meta-analysis, which works suggest the absolute omission of (early) adjuvant RT. The authors debate this general ascertainment and provide a kind of a „critical appraisal” of this statement. This approach, the criticism of the findings above, and the particular analysis of the biological/medical/statistical factors in the background is really important to understand/optimize the best care of our patients. To summarize the reviewer’s opinion, the material is worthy to consider for publication. However, the reviewer has some notifications/concerns/recommendations.

1, It would be better to use consequently „early salvage RT” (eSRT) alongside the material to distinguish it from the other forms of salvage RT.

Ad 1.: Thank you for this recommendation. This was corrected throughout manuscript.

2, Consider the use of „postoperative” (e.g. in the title) since sometimes (e)SRT is carried out 2-3 years after the surgery intervention.

Ad 2.: We chose not to change the term to postoperative (RT) since this is a term incorporating both adjuvant AND salvage RT, whereas salvage radiotherapy is a fix term describing a treatment of usually 66-70 Gy in case of persisting PSA after RP or PSA rise after RP (as opposed to adjuvant RT of usually 60 Gy after an adequate fall of postoperative PSA). We would prefer to keep this term in the manuscript. 

3, The ARTISTIC meta-analysis specified some weaknesses of the work as well, e.g. the short follow-up periods, the low number of events, the incalculable effect of endocrine treatment, the overrated value of PSA follow up etc. It would be worthy to mention it.

Ad 3.: Thank you for this notification: The fact that the authors themselves mentioned the (possible) weaknesses of the work was included in the manuscript in the conclusion:

“These new data greatly improve patient selection towards eSRT in patients without multiple high-risk features and led to a reduction of “unnecessary” adjuvant treatments. However, given the above-mentioned limitations, several of which having also been voiced by the authors of the RCTs and the metaanalysis, the interdisciplinary working group within the German S3 guideline consortium has agreed upon recommendations for patient counselling in specific clinical scenarios where ART still appears to be a valid treatment option (Figure 2).”

4, The ARTISTIC meta-analysis excluded the obligate effect of different prognostic factors, like basic PSA level, Gleason score, R state, vesicle seminalis infiltration, CAPRA score etc. It would be interesting to interpret it.

  1. Ad 4.: There are (pre-defined) subgroup analyses of EFS sorted by study and by the above-mentioned risk factors in the metaanalysis. However, the power of these subgroup analyses is limited by the low event rate and the low number of patients in these subgroups.

5, It is recommended to mention the significance of CAPRA score and Decipher molecular assay in the Discussion/Conclusion part.

Ad 5.: The CAPRA-S score (consisting of presurgical PSA level, Gleason Score, R status, ECE status, seminal vesical invasion and lymph node invasion) is no doubt of high clinical relevance for the decision making considering a post-operative radiotherapy. It consists of the (above-mentioned) post-operative risk factors GS, R status, seminal vesicle infiltration and ECE which have been discussed in the manuscript. The only additional pre-surgical factor (initial PSA) did not influence the decision for eSRt or ART in these analyses and we therefore chose not to speculate on its influence.

6, The 3 RCT differed from each other considering inclusion criteria as well, e.g. in the obligatory number of risk factors (e.g. GETUG-AFU 17 required 3 risk factors, and interestingly in this study there was some trend in favor of ART, see ARTISTIC manuscript). It would be interesting to interpret it.

Ad 6.: Thank You very much for this important point. In Table and manuscript, these differences have been pointed out. We believe that the points of criticism in the discussion and the conclusion of our work already consider these factors well.

7, Abstract, lines 27-28. It is recommended to shorten/close up the „can be offered” categories. 

Ad 7.: The sentence has been rephrased as follows:

“ART should be offered in case of pT3/pT4 and R1 and Gleason-Score 8-10; ART can be offered in case of pT3/pT4 and R0 and Gleason 8-10 as well as in case of multifocal R1 (including pT2) and Gleason 8-10”

8, Line 41: The PSA monitoring was really „inadequate” in these studies?

Ad 8.: The adjective was exchanged to “imprecise”: At the time of the mentioned RCTs, PSA sensitivity reached a minimum of 0.1 ng/ml while currently, thresholds of 0.01 ng/ml can be routinely detected through blood tests. 

9, Comments: „Immortal time bias„ could modify the results only some patients. Some delay of referral to SRT (line 261) likely has only minimal general effect as well.

Ad 9.: This comment is correct: The magnitude of the “immortal time bias” effect is unclear. We believe that it is nevertheless worth mentioning in this specific study design as a possible bias due to the fact that all three RCTs started “counting” late toxicities and oncological outcome parameters at the time point of randomization.

10, It is interesting to discuss the role of PSMA-PET in this setting.

Ad 10.: We have deliberately chosen not to “dive into” the possibilities of more recent diagnostic options such as PSMA-PET-CT (or -MR) or Choline-PET. The investigated prospective data was performed without stringent use of PET-imaging. Consequently, all comparisons to more recent data of PSMA-PET in the post-operative setting remains speculative and further RCTs are needed for a valid estimation of their benefit. Secondly, whether recurrences after ART or SRT occur due to already existing (detectable) metastases prior to RT or occur “in field” is a completely separate question and cannot be answered based on the investigated data.

Reviewer 2 Report

In the communication entitled “ Postoperative Radiotherapy of Prostate Cancer: Adjuvant versus Salvage” the authors analyzed in detail recent trials that contribute with convincing evidence to the decision to treat post-surgery prostate cancer patients with salvage radiotherapy.

In general, the manuscript is well written and organized, with an adequate bibliographic review that supports the discussion and conclusions reached by the authors. However, consistent with the authors' statement in the paragraph: "More clinical data appear to be needed on the optimal management of patients after RP. For example, on the optimal use of ADTs (eg, RADICALS-HD), new radiotherapy technologies, on the role of hypofractionation, on the use of PET-CT and on the use of biomarkers for better risk stratification”, I suggest that it would be interesting to discuss in the manuscript the possibilities and advantages of targeted radiotherapy, such as the recently FDA-approved 177Lu-PSMA-617 in cancer patients after RP.

Author Response

In the communication entitled “ Postoperative Radiotherapy of Prostate Cancer: Adjuvant versus Salvage” the authors analyzed in detail recent trials that contribute with convincing evidence to the decision to treat post-surgery prostate cancer patients with salvage radiotherapy. 

In general, the manuscript is well written and organized, with an adequate bibliographic review that supports the discussion and conclusions reached by the authors. However, consistent with the authors' statement in the paragraph: "More clinical data appear to be needed on the optimal management of patients after RP. For example, on the optimal use of ADTs (eg, RADICALS-HD), new radiotherapy technologies, on the role of hypofractionation, on the use of PET-CT and on the use of biomarkers for better risk stratification”, I suggest that it would be interesting to discuss in the manuscript the possibilities and advantages of targeted radiotherapy, such as the recently FDA-approved 177Lu-PSMA-617 in cancer patients after RP.

Ad Reviewer #2:

Thank you for your evaluation of our manuscript. The upcoming targeted therapies indeed show great potential for improvement of outcomes of prostate cancer patients, possibly also in the post-operative setting. However, we believe that a discussion of targeted therapies in the setting of postoperative radiotherapy should not be part of this manuscript for the following reasons: The aim of this manuscript is to evaluate the (only) two current therapies which both offer a chance for curation after RP as has been shown through several RCTs with long-term follow-up. Targeted therapies such as 177Lu-PSMA-617 are only approved in a metastatic setting and do not offer a curative approach. There is also no prospective data of RCTs of these therapies in this setting.
